# Starvation-Induced Differential Virotherapy Using an Oncolytic Measles Vaccine Virus

**DOI:** 10.3390/v11070614

**Published:** 2019-07-05

**Authors:** Gabriel Scheubeck, Susanne Berchtold, Irina Smirnow, Andrea Schenk, Julia Beil, Ulrich M. Lauer

**Affiliations:** 1Department of Internal Medicine VIII, University Hospital Tuebingen, Otfried-Mueller-Strasse 10, D-72076 Tuebingen, Germany; 2German Cancer Consortium (DKTK), DKFZ Partner Site Tuebingen, Otfried-Mueller-Strasse 10, D-72076 Tuebingen, Germany

**Keywords:** starvation, fasting, virotherapy, oncolysis, measles vaccine virus

## Abstract

Starvation sensitizes tumor cells to chemotherapy while protecting normal cells at the same time, a phenomenon defined as differential stress resistance. In this study, we analyzed if starvation would also increase the oncolytic potential of an oncolytic measles vaccine virus (MeV-GFP) while protecting normal cells against off-target lysis. Human colorectal carcinoma (CRC) cell lines as well as human normal colon cell lines were subjected to various starvation regimes and infected with MeV-GFP. The applied fasting regimes were either short-term (24 h pre-infection) or long-term (24 h pre- plus 96 h post-infection). Cell-killing features of (i) virotherapy, (ii) starvation, as well as (iii) the combination of both were analyzed by cell viability assays and virus growth curves. Remarkably, while long-term low-serum, standard glucose starvation potentiated the efficacy of MeV-mediated cell killing in CRC cells, it was found to be decreased in normal colon cells. Interestingly, viral replication of MeV-GFP in CRC cells was decreased in long-term-starved cells and increased after short-term low-glucose, low-serum starvation. In conclusion, starvation-based virotherapy has the potential to differentially enhance MeV-mediated oncolysis in the context of CRC cancer patients while protecting normal colon cells from unwanted off-target effects.

## 1. Introduction

Starvation (fasting) has been shown to sensitize malignant cells while protecting normal cells at the same time [1]. In cancer research, this phenomenon is termed differential stress resistance (DSR) and has recently moved into focus given its potential for widespread clinical applications [2,3]. In the course of gaining numerous mutations, most tumor cells achieve the ability to proliferate independently of external growth factors and are thus prone to becoming more vulnerable to stress conditions, like chemotherapy, when deprived of nutrients [1,2]. In this context, the insulin-like growth factor-1 (IGF-1) pathway has been proposed to be growth-promoting and anti-apoptotic, thereby favoring carcinogenesis [3]. Fasting, which is defined as a short-term transient total absence of food, has shown evidence of being more efficient in reducing levels of IGF-1 and glucose than any prolonged caloric restriction and, furthermore, favors chronic weight loss that is highly deleterious to most cancer patients [4].

A recent study by Lee et al. [5] revealed that 15 out of 17 mammalian cancer cell lines could be sensitized to doxorubicin and cyclophosphamide (CP) by fasting 24 h prior to and 24 h post-chemotherapy. Next, when using a mouse tumor model, it could be shown that two cycles of fasting were as effective in the treatment of human metastatic cancer as two cycles of chemotherapy with CP. Interestingly, the combination of both, i.e., starvation plus chemotherapy, had been shown to be the most effective regimen, which suggests that fasting has the potential to increase the efficacy of chemotherapy [5] and potentially, also, of other non-genotoxic forms of cancer therapies. Even more puzzling, the starvation of tumor-bearing mice allowed researchers to increase the administered doses of chemotherapy with etoposide (ETO) up to three times of the maximal dose approved in humans [6]. Whereas ETO at this concentration killed 43% of control mice, only 6% of the mice that were pre-starved died after ETO treatment [6].

To date, only a small number of clinical trials have explored the effect of combined fasting and chemotherapy in patients [7,8,9,10]. A case report on 10 patients suffering from different cancer entities showed a significant reduction of side effects, such as fatigue or weakness, when fasting was performed 48–140 h prior to and 5–56 h post-chemotherapy [7]. Another more recent clinical study evaluated the effects of short-term fasting on tolerance to adjuvant chemotherapy in HER-2-negative breast cancer patients [9]. As a result, erythrocyte and thrombocyte counts post-chemotherapy were higher in fasted patients. In 34 women with breast and ovarian cancer, 60 h fasting plus chemotherapy not only proved to be safe and feasible, but also improved the quality of life, well-being, and fatigue when compared to chemotherapy alone [10]. Accordingly, fasting was well-tolerated in cancer patients and has the potential to reduce side effects, but there is a lack of data supporting that it can increase the efficiency of current anticancer therapies, as data from in vitro and animal studies has demonstrated.

Many larger clinical trials are currently ongoing to determine possible benefits of fasting regarding efficacy of treatment, adverse events, quality of life, weight changes, and changes in metabolic, hormone, and inflammatory response (NCT00936364, NCT01802346, NCT02710721, NCT03162289, NCT03340935, NCT03595540, NCT03709147, NCT03700437, NCT01175837; please note that the cited clinical studies are denoted by their ClinicalTrials.gov identifiers). In an ongoing study at the University Hospital Tübingen (NCT02607826), short-term starvation for a timeframe beginning 24 h prior to chemotherapy administration and lasting until 6 h after administration is being tested in patients suffering from a wide variety of solid tumors.

Oncolytic virotherapy has recently gained attention due to demonstration of the first cancer patient who experienced a complete remission that has been ongoing for more than five years now following treatment with a single high-dose shot of recombinant oncolytic measles vaccine virus expressing the sodium iodide symporter (MeV-NIS) [11]. Hopes that this will not remain an isolated case have been encouraged by the recent approval of IMLYGIC™ (T-Vec; talimogene laherparepvec), the first oncolytic immunotherapeutic treatment for unresectable skin and lymph node lesions in patients with advanced melanoma [12]. Numerous clinical trials have assessed the safety and effectiveness of oncolytic viruses [13,14].

ONCOS-102, a granulocyte-macrophage colony-stimulation factor (GM-CSF)-encoding oncolytic adenovirus, has been demonstrated to induce infiltration of CD8^+^ T-lymphocytes into initially T-cell-negative mesothelioma in a 68-year-old patient [15]. Accordingly, also poorly immunogenic tumors could be sensitized to immunotherapy by oncolytic viruses (OVs), which open up new possibilities for T-cell-based approaches in cancer treatment.

Reovirus is another OV being evaluated in numerous tumor entities in phase II clinical trials in combination with conventional chemotherapy [16,17,18,19,20,21]. REOLYSIN^®^, an isolate of reovirus type 3, was well-tolerated in patients, but did not improve progression-free survival (PFS) in metastatic colorectal carcinoma, non-small cell lung cancer (NSCLC), breast cancer, and prostate cancer compared to standard treatment arms; however, it was demonstrated to extend overall specific survival (OSS) specifically in breast cancer patients [16,17,20,21]. Reovirus is currently being investigated in a phase I trial in combination with GM-CSF for the treatment of high-grade relapsed or refractory brain tumors (NCT02444546).

Also, the use of recombinant tumor-targeting viruses, such as measles vaccine virus (MeV), has been demonstrated to be a safe and highly promising approach in cancer treatment [22,23,24,25,26,27,28,29]. However, widespread resistance to oncolysis following MeV inoculation was observed when challenging the NCI-60 tumor cell panel with oncolytic MeV [30]. This might be an important phenomenon that currently hinders the broader clinical success of virotherapy with basic, non-optimized, first-generation vector types.

Until now, only one study, published by Esaki et al., has investigated whether starvation may enhance the effectiveness of OVs [31]. In this study, fasting was demonstrated to increase the replication and oncolytic activity of oncolytic herpes simplex virus (oHSV) in glioblastoma multiforme (GBM) cells, but not in human astrocytes. These results were confirmed in vivo, showing enhanced virus replication in starved mice [31].

Oncogenes that prevent the starvation-induced cellular switch to a protected mode render the tumor cell vulnerable to stress such as chemotherapy and potentially also OVs. Consequently, this prompted our hypothesis that DSR could be a way to sensitize colorectal tumor cells to virotherapeutics while making healthy cells more robust against virus-mediated oncolysis.

## 2. Materials and Methods

### 2.1. Cell Culture

Human colorectal cancer cell lines HT-29, HCT-15, and HCT-116 are cell lines from the U.S. National Cancer Institute’s NCI-60 tumor cell panel and were purchased from Charles River Laboratories (Charles River Laboratories Inc., New York, NY, USA). Normal human cell lines CCD-18 Co (non-malignant fibroblastic colon cells) and CCD-841 CoN (non-malignant epithelial colon cells) were obtained from the American Type Culture Collection (ATCC, Manassas, VA, USA). African green monkey kidney (Vero) cells were obtained from the German Collection of Microorganisms and Cell Cultures (DSMZ, Braunschweig, Germany).

Tumor cells and Vero cells were maintained in high-glucose (4.5 g glucose per liter) Dulbecco’s Modified Eagle’s Medium (DMEM, D6429, Sigma-Aldrich, St. Louis, MO, USA) supplemented with 10% fetal calf serum (FCS; Biochrom, Berlin, Germany). Normal human cell lines were grown in Alpha Modified Eagle’s Medium (Alpha-MEM, BE12-169F, Lonza, Verviers, Belgium) enriched with 10% FCS and 10 mM L-glutamine (Gibco, Paisley, Scotland, UK). Cell culture flasks were stored in humidified incubators at 37 °C under 5% CO_2_.

### 2.2. Experimental Setup

Cells were seeded in 24-well plates (TPP, Trasadingen, Switzerland) in 0.5 mL standard medium/well (see below) (HCT-15: 3 × 10^4^; HCT-116: 2.5 × 10^4^; HT-29: 4 × 10^4^; CCD-18 Co: 2 × 10^4^; CCD-841 CoN: 2 × 10^4^ cells/well) at day 0. Medium either remained in the well (see following point 1) or was switched to 0.5 mL starvation medium/well (see following point 2) at day 1. After infection at day 2, cells were grown in standard or starvation medium until the endpoint assay was performed.

Standard cell culture medium:

For tumor cells HT-29, HCT-15, and HCT-116 high-glucose DMEM (containing 4.5 g glucose per liter) supplemented with 10% FCS was used. For non-malignant colon cells, CCD-18 Co and CCD-841 CoN Alpha-MEM (containing 1 g glucose per liter) supplemented with 10% FCS was used.

1 Standard regimen (no starvation)

Determination of cell line-specific multiplicity of infection (MOI) resulting in approximately 25% oncolysis rate: Cells were seeded in standard medium at day 0. After infection with ascending MOIs at day 2 (see Section 2.4), medium was replaced with standard medium. Remnant cell mass was determined by the sulforhodamine B (SRB) assay at 96 h post-infection (hpi) at day 6.

2 Starvation regimens

Short-term low-glucose, low-serum starvation (=short-term starvation): Cells were cultured in serum- and glucose-restricted culture medium for 24 h before infection. At day 1, standard medium was replaced with low-glucose, low-serum medium (glucose concentrations of 0, 0.5, 1, 2, and 4.5 g/L supplemented with 1% FCS each; control: standard medium). Starvation medium was prepared by mixing high-glucose DMEM (D6429, Sigma-Aldrich, St. Louis, MO, USA) + 1% FCS and glucose-free DMEM (F0405, Biochrom, Berlin, Germany) + 1% FCS. At day 2, medium was removed, replaced with infection medium (see Section 2.4), and finally switched to standard medium in which cells were cultivated until the endpoint assay was performed (24 h starvation period).

Long-term low-glucose, low-serum starvation: Cells were cultured in serum and glucose-restricted culture medium for 24 h before and 96 h post-infection. Medium was switched to low glucose, low serum at day 1. After infection at day 2, the starvation medium was renewed until read-out (120 h starvation period).

Long-term low-glucose, standard serum starvation: Cells were cultured in glucose-restricted culture medium (standard serum culture medium) for 24 h before infection and 96 h post-infection. Low-glucose, standard serum medium (glucose concentrations of 0, 0.5, 1, 2, and 3 g/L supplemented with 10% FCS; control: standard medium) was applied at day 1. After infection at day 2, the starvation medium was renewed (120 h starvation period).

Long-term standard glucose, low-serum starvation: Cells were cultured in serum-restricted cell culture medium (standard glucose culture medium) for 24 h before and 96 h post-infection. Standard glucose, low-serum medium (glucose concentration of 4.5 g/L (tumor cells) and 1 g/L (normal colon cells), supplemented with 1%, 2.5%, 5%, or 7.5% FCS; control: standard medium) was applied at day 1. After infection at day 2, the starvation medium was renewed (120 h starvation period).

### 2.3. Oncolytic Virotherapeutic MeV-GFP

A commercially available original monovalent vaccine batch of measles virus (MeV) strain Mérieux (Sanofi-Pasteur, Leimen, Germany) was modified by insertion of a gene encoding green fluorescent protein (MeV-GFP) as previously described [32].

### 2.4. Virus Infection

Infection was performed at indicated multiplicities of infection (MOI) at day 2. MOCK-treated cells (no virus, infection medium only) served as control. Cells were washed once with PBS and then treated with a distinct MOI of MeV-GFP diluted in 250 µL Opti-MEM (Opti-MEM + GlutaMAX Supplement, Gibco, Paisley, Scotland, UK). At 3 h post-infection (hpi), the inoculum was removed and changed to standard or starvation medium.

### 2.5. Sulforhodamine B (SRB) Cell Viability Assay

Cells were seeded in 24-well plates and infected 48 h later. At 96 hpi, cells were washed with ice-cold PBS, fixed with 10% trichloroacetic acid (TCA), and incubated at 4 °C for 30 min. Then, TCA was removed, and the fixed cells were washed with tap water and dried for 24 h. To stain cells, 250 µL of SRB dye (0.4% dissolved in 1% acetic acid) was added followed by incubation at RT for 10 min, and cells were washed with 1% acetic acid and dried again. Protein-bound dye was dissolved in 10 mM Tris base for 10 min before measurement of optical density was performed in a 96-well microtiter plate reader (Tecan Genios Plus, Tecan Deutschland, Crailsheim, Germany) at a wavelength of 550 nm (reference wavelength of 620 nm). The SRB assay allows densitometric quantification of the total cellular protein mass after incubation of the cell mass of interest with cytotoxic substances for a distinct time range; thereby, a remnant cell mass is calculated which represents the cytotoxic effectiveness measured under the respective experimental conditions.

### 2.6. Lactate Dehydrogenase (LDH) Assay

Cells were seeded in 24-well plates, starved starting from day 1, and infected with MeV-GFP at day 2 using the indicated MOIs. Values corresponding to medium without cells were subtracted as the blank for each starving condition. At 96 hpi, cell culture supernatant was transferred into new plates and cells were lysed in 0.1% Triton X100 (dissolved in PBS) (Sigma-Aldrich, St. Louis, MO, USA) for 10 min. Supernatant, lysate, and medium without cells (10 µL each) were analyzed in a 96-well plate. The enzymatic reaction was started by adding 200 µL pyruvate/NADH reagent (LDH-Kit, Analyticon Biotechnologies, Lichtenfels, Germany). LDH release was quantified by photometric measurement of NADH decrease in a microtiter plate reader (Magellan, Tecan Deutschland, Crailsheim, Germany) at a wavelength of 340 nm.

The percentage of cell lysis was calculated as follows:(1)$$\%\;\mathrm{cell}\;\mathrm{lysis}\;=\;\frac{\mathrm{LDH}\;\left[\mathrm{supernatant}\right]\;-\;\mathrm{LDH}\;[\mathrm{medium}]}{\mathrm{LDH}\;\left[\mathrm{lysate}\right]\;+\;(\mathrm{LDH}\;\left[\mathrm{supernatant}\right]\;-\;\mathrm{LDH}\;[\mathrm{medium}])}\;\times \;100$$

The LDH assay detects lactate dehydrogenase (LDH) activity, an enzyme that is released by loss of cell membrane integrity due to necrosis or toxic cell damage. It serves as a standard parameter for cell lysis taking place under the respective experimental conditions.

### 2.7. Viral Growth Curve

A total of 3 × 10^5^ cells/well were seeded in a 6-well plate, starvation medium was applied at day 1 (1 mL/well), and MeV-GFP infection (MOI as indicated) was performed in 1 mL Opti-MEM at day 2. At 3 hpi, the infection medium was removed, and cells were washed three times with PBS before standard or starvation medium was renewed. Supernatants and cell lysates (scraped into 1 mL Opti-MEM) were harvested at 3, 24, 48, 72, and 96 hpi and subjected to one freeze–thaw cycle. Viral titers of supernatants and cell lysates were quantified by fluorescence microscopy of GFP on Vero cells and then calculated as total viral titer (supernatants + lysates).

In detail, Vero cells (5 × 10^4^ cells/mL) were seeded in DMEM + 5% FCS in 96-well plates (200 µl/well). The following day, samples were thawed, vortexed, and centrifuged (2 min, 3000 rpm, 22 °C). Supernatants were used to prepare a dilution series at concentrations ranging from 1 × 10^0^ to 1 × 10^−7^ in DMEM + 5% FCS of which 50 µL each were transferred onto Vero cells. At 96 hpi, fluorescence microscopy of GFP expression (IX50, Olympus, Tokyo, Japan; analySIS, Soft Imaging System, Münster, Germany) allowed for the calculation of viral titers according to the method of Kärber and Spearman [33,34].

### 2.8. Statistics

The results of cell viability assays and viral growth curves are expressed as mean ± standard deviation (SD). Graphics were created using GraphPad Prism Software version 4.01 (GraphPad Software, La Jolla, CA, USA). Statistical analysis was conducted with JMP Software version 12.2.0 (SAS Institute Inc., Cary, NC, USA).

In order to make a statement about the influence of starvation on virus-mediated oncolysis, the ratios of uninfected (MOCK) versus virus test groups (VIRUS) were compared based on whether these groups were cultured under starving or standard medium conditions. For this purpose, the differences of logarithmized MOCK and VIRUS group cell mass values (quotient of non-logarithmized values) (log_10_ MOCK−log_10_ VIRUS) were compared between starved groups and control groups for each run of the experiment by the Dunnett’s multiple comparison test. *P*-values < 0.05 were considered as statistically significant.

For illustration purposes, we delogarithmized the data and termed the results quotient of geometric mean (QoGM).

(2)$$\mathrm{QoGM}\;=\;10^{\left(log_{10}\;MOCK\right)-\left(log_{10}\;VIRUS\right)}$$

QoGM serves as a parameter for virotherapy efficacy. High values of QoGM in relation to control (no starvation) indicate an increase of virus-mediated oncolysis efficacy, whereas low values indicate a decrease.

## 3. Results

To comprehensively investigate starvation-induced differential virotherapy using an oncolytic measles vaccine virus, our study comprised a two-step approach for each cell line. At first, cells were grown in standard medium and treated with different virus concentrations to determine MOIs resulting in a “moderate” oncolysis of around 25%, leading to a cell line-dependent adjustment of the respective virus dosages for each cell line. These MOIs then were used in subsequent experiments applying different starvation regimens investigating a combined starvation-based virotherapy. Cells were either starved for 24 h pre-infection, termed short-term starvation (Figure 1a) or starved 24 h pre-infection and 96 h post-infection (Figure 1b). For all regimens, cell viability assays were performed at 96 hpi.

### 3.1. Determination of Adjusted MOIs in Colon Carcinoma and Non-Malignant Colon Cell Lines

First, we set out to analyze distinct malignant and non-malignant colon cells by SRB viability assays in an individual manner, in order to identify the distinct MOI leading to a cell mass reduction of about 25% (Figure 2). Then, these cell line-adapted MOIs were used to investigate the influence of starvation on virotherapy-induced cytotoxicity in further experiments (please note that these suboptimal cell-killing MOIs were chosen in order to provide enough “space” to measure any additional cell-killing effects of virus-mediated oncolysis that were a result of starvation).

Based on this experimental design, we treated both malignant and non-malignant colon cell lines with ascending MOIs with the virotherapeutic vector MeV-GFP. Remaining cell masses were determined by SRB viability assays at 96 hpi. For colorectal cancer cell line HCT-15 (Figure 2c), only MOI 10 led to a marginal cell mass reduction of 17% (please note that we did not apply MOIs higher than 10 since it is clinically challenging to achieve such virus concentrations in cancer patients). By contrast, human colon cancer cell lines HCT-116 (Figure 2b, MOI 0.75, cell mass reduction 33%) and HT-29 (Figure 2a, MOI 0.5, cell mass reduction 39%) were found to be more susceptible to MeV-GFP-mediated oncolysis.

MeV-GFP infection of non-malignant CCD-18-Co cells (Figure 2d), a human colon fibroblast cell line (MOI 10, cell mass reduction 29%), and of non-malignant CCD-841-CoN cells (Figure 2e), a human colon epithelial cell line (MOI 5, cell mass reduction 33%), showed patterns of virotherapy resistance similar to HCT-15.

### 3.2. Short-Term Starvation (24 h) Decelerated Tumor Cell Growth and Kept MeV-GFP-Mediated Oncolysis Intact

Next, combined starvation plus viral oncolysis settings were investigated. For this purpose, human colon carcinoma cell lines HT-29 (Figure 3a), HCT-15 (Figure 3c), and HCT-116 (Figure 3e) were starved for 24 h pre-infection in low-glucose, low-serum medium. Then, these pre-starved malignant cells were either infected with the above determined MOIs of MeV-GFP or MOCK-infected. In each experiment, control group data (blue frames) are shown from cells that were cultivated in standard (non-starved) cell culture medium. Remnant cell mass was determined by SRB assay at 96 hpi and is illustrated using bar graphs.

Remarkably and consistently for all cell lines, even a short period of 24 h “starvation only” (no infection) was found to be sufficient to reduce the tumor cell mass by 4% (4.5 g Glc/L, 1% FCS) up to ~10% (0 g Glc/L, 1% FCS). Accordingly, we observed a tendency for greater inhibition of tumor cell growth in cell lines HT-29 and HCT-15 the lower the levels of glucose and serum were (Figure 3a,c). The extent of “starvation only”-induced cell mass reduction was found to be equal throughout all cell lines and independent of any virotherapy resistance characteristics.

To better illustrate the efficacy of virus-mediated oncolysis under starvation conditions, we compared the ratios of uninfected (MOCK) and infected (VIRUS) cell mass groups between starving and control groups. Results are reported as “quotient of geometric mean” (QoGM) functioning as an indicator for virotherapy efficacy and are displayed in scatter dotplots (for details, see Section 2.8). For HT-29 cells, we observed a trend of increasing QoGM at 0 g Glc/L and 1% FCS, however, values of QoGM were not found to significantly differ between starving and non-starving control groups (Figure 3b). In summary, the efficacy of MeV-mediated oncolysis in HCT-15 and HCT-116 cells remained predominantly unaffected by short-term starvation, while in HT-29, an additional effect of starvation plus virotherapy could be observed.

### 3.3. Long-Term Starvation Substantially Inhibited Tumor Cell Growth and Enhanced the Efficacy of MeV-GFP-Mediated Oncolysis for HT-29 Cells Cultured in Low-Glucose, Low-Serum But Not in Low-Glucose, Standard Serum Medium

To further intensify cytotoxic effects, we extended the fasting period to a total timespan of 120 h (Figure 4). For this purpose, HCT-15, HCT-116, and HT-29 tumor cells were starved 24 h pre-infection and 96 h post-infection either in low-glucose, low-serum medium (Figure 4a,c,e) or at low-glucose, standard serum medium (Figure 4b,d,f). Cells were either infected with MeV-GFP or MOCK-infected before remnant cell masses were determined by SRB assay at 96 hpi.

As shown by bar graphs, long-term “starvation only” (no infection) dramatically reduced tumor cell masses of all cell lines (black bars). Combining starvation with MeV-GFP treatment (checkered bars) resulted in measles virus-induced cell killing that was still effective: For HCT-15 cells, treatment with MeV-GFP led to a cell mass reduction of ~22% under standard medium conditions (control) and to a proportional extent either during low-glucose, low-serum (Figure 4c), or low-glucose, standard serum starvation conditions (Figure 4d). The same applies to HCT-116 cells, where a 35% cell mass reduction under control conditions was similarly observed under starvation conditions (Figure 4e). Exceptions apply to the low concentration ranges, especially in low-glucose, low-serum starvation, where remnant cell masses were nearly zero, and therefore no remarkable difference between “starved only” and “starved plus virotherapy” groups could be measured.

For HT-29 cells, the virus-mediated cell-killing rate under standard conditions correlates to the extent of low-glucose, standard serum conditions (Figure 4b). Notably, MeV-mediated oncolysis rate doubled under low-glucose, low-serum starvation (4.5 g Glc/L, 1% FCS and 2 g Glc/L, 1% FCS), compared to control (Figure 4a). These findings were confirmed when looking at our parameter QoGM indicating virotherapy efficacy: “QoGM” increased significantly for HT-29 cells when long-term starvation conditions were applied (at 4.5 g and 2 g Glc/L, 1% FCS compared to control). The decrease of QoGM at 0 g Glc/L, 1% FCS can be attributed to the very low amount of remaining cell mass, where no significant differences between infected and uninfected groups could be measured. In summary, low-glucose, low-serum starvation has the potential to enhance MeV-mediated oncolysis efficacy in HT-29 CRC cells, in terms of a synergistic effect.

In HCT-15 and HCT-116 cells, values of QoGM stayed unchanged after long-term low-glucose, low-serum, as well as after low-glucose, standard serum starvation (Figure 4c–f), meaning that under long-term starvation, MeV-GFP-mediated oncolysis worked to the same extent as under standard conditions. Consequently, an additional cell-killing effect of starvation plus virotherapy could be observed for long-term starvation in HCT-15, HCT-116, and HT-29 CRC cells (glucose only starvation).

### 3.4. MeV-GFP Replication was Impaired by Long-Term Low-Glucose, Lo-Serum Starvation, But Widely Unaffected by Long-Term Standard Glucose, Low-Serum Starvation, and Increased in Short-Term Low-Glucose, Low-Serum-Starved HT-29 Cells

Since our results indicate that starvation decelerates tumor cell growth and may even increase the efficacy of MeV-GFP-mediated oncolysis in HT-29 cells, we wanted to investigate how starvation affects virus replication.

For this purpose, we generated virus growth curves (Figure 5). HT-29 cells were starved either 24 h pre-infection (short-term starvation) or 24 h pre-infection and 3–96 h post-infection (long-term starvation) in low-glucose, low-serum (red and green curves) or standard glucose, low-serum medium (black curve). A respective blue curve depicts the control group where cells were grown under standard conditions (no starvation). Infection with MeV-GFP was performed at an MOI of 0.5. Titration of HT-29 cell supernatants and lysates on recipient Vero cells delivered virus titers as plaque-forming units (PFU)/mL.

Short-term starvation (Figure 5a) at 0 g Glc/L, 1% FCS increased the replication of oncolytic MeV-GFP in HT-29 cells. By contrast, short-term starvation at higher glucose concentrations did not alter viral replication compared to control.

When cells were continuously starved after infection (long-term starvation, Figure 5b), virus replication was roughly proportionally diminished to the intensity of starvation, except for one outlier at 2 g Glc/L, 1% FCS at 72 hpi. In detail, viral replication was notably lowered at glucose and serum restriction (red and green curve). Under serum restriction only (black curve), viral replication followed a similar pattern to that of not starved cells (blue curve) and exhibited that viral replication is widely unaffected by long-term standard glucose, low-serum starvation.

### 3.5. Long-Term Serum Starvation Increased MeV-GFP-Mediated Oncolysis in Human Colon Carcinoma HT-29 Cells, But Not in Normal Human Colon Cells CCD-18 Co and CCD-841 CoN

Since our results revealed that especially long-term serum restriction favors MeV-GFP-mediated oncolysis in distinct cancer cells and, at the same time, impairs measles vaccine virus replication only marginally, we set out to investigate the effects of serum starvation more deeply. Further, we were interested in how normal human colon cells respond to nutrient restriction and virotherapy. As illustrated in Figure 6, CCD-18 Co (human colon fibroblast cell line, Figure 6a), CCD-841 CoN (human colon epithelial cell line, Figure 6b) and HT-29 (human colorectal carcinoma cell line, Figure 6c) cells underwent long-term standard glucose, low-serum starvation. Cell lines were infected with MeV-GFP (checkered bars) or MOCK-infected (black bars), respectively, and remnant cell masses were determined by SRB assay.

“Starvation only” reduced cell masses in non-malignant cells CCD-18 Co (Figure 6a) and in non-malignant CCD-841 CoN cells (Figure 6b) under descending FCS concentrations. In malignant HT-29 cells, lower concentrations of serum led to reduction of cell mass (Figure 6c). Note that the standard medium for normal colon cells contains only 1 g Glc/L, whereas the standard medium for neoplastic cells contains 4.5 g Glc/L. Additional MeV-GFP treatment enhanced cell killing in HT-29 at all serum concentrations, with up to 89% cell mass reduction for combined starvation and virotherapy. MeV-GFP-infection of CCD-18 Co and CCD-841 CoN cells reduced cell masses to a small extent at standard serum levels, but did not lead to increased cell killing under descending FCS concentrations.

However, it is important to mention that non-malignant colon cells were infected with a high MOI of either 10 or 5 (in contrast to MOI 0.5 which was used for HT-29 cancer cell infection). This was done in order to compensate for the much lower expression of the measles vaccine virus receptor (i.e., CD46) on all non-malignant cells.

As illustrated by scatter dotplot, values of QoGM indicate that MeV-mediated oncolysis efficacy in starved HT-29 cells was increased by up to three-fold at 1% FCS (*p* = 0.023) (Figure 6c). By contrast, starvation impaired virus-induced cell killing significantly in CCD-18 Co cells (Figure 6a) and slightly in CCD-841 CoN cells (Figure 6b).

To distinguish whether cell mass reduction was caused by (i) inhibition of cell proliferation or (ii) direct cell lysis, LDH release was quantified as a marker of direct cell lysis (Figure 7a–c). Values of “starvation only”-induced cell lysis (black bars) were at a modest level and only rose slightly with increasing starvation intensity at a range of 7%–12% for CCD-18 Co (Figure 7a), 8%–20% for CCD-841 CoN cells (Figure 7b), and 13%–20% for HT-29 (Figure 7c). By contrast, infection of HT-29 cells with MeV-GFP (MOI 0.5) under serum restriction (checkered bars) approximately doubled the lysis rate compared to standard conditions (37%–68%). For CCD-18 Co and CCD-841 CoN cells, only a moderate increase was found after MeV-GFP infection. Taken together, our QoGM parameter for virotherapy efficacy showed an increase of cell lysis efficacy for HT-29 cells (*p* = 0.010), whereas QoGM remained unchanged for non-malignant CCD-18 Co and CCD-841 CoN cells.

## 4. Discussion

Even though much progress has been made in the prevention, screening, and treatment of colorectal carcinoma (CRC), it still remains today’s third most common cause of cancer-related deaths worldwide [35].

Oncolytic virotherapy as an alternative treatment option is currently being investigated for various malignancies. Effective OVs can selectively infect, replicate in, and lyse cancer cells where effective antiviral defense mechanisms are compromised due to various genetic mutations [36]. In addition to direct cell lysis, OVs may initiate a profound and long-lasting antitumoral immunogenicity [37,38]. Given that nutritional depletion had been shown to modulate nutrient signaling pathways, sensitize cancer cells to chemotherapeutics, and protect normal cells [6], we sought to investigate the effects of nutrient restriction on oncolytic virotherapy with the virotherapeutic vector MeV-GFP.

In the present study, we found that long-term starvation is capable of enhancing the oncolytic potential of MeV-GFP specifically in the human colon cancer cell line HT-29. Under standard conditions, all cell lines were lysed by our vector MeV-GFP, and the extent correlated with the employed MOI. We initially tested the impact of short-term starvation on virus-mediated cell killing. Colon cancer cells deprived of glucose and serum for 24 h pre-infection were reduced by up to 10% in cell mass. Infection with our vector MeV-GFP further reduced tumor cell mass, however, without potentiating the effect. As expected, when the fasting period was extended to 24 h pre- and 96 h post-infection, cell masses were more dramatically reduced. Interestingly, our results delivered evidence that serum restriction in HT-29 cells enhanced the efficacy of MeV-GFP-mediated oncolysis, whereas a restriction in glucose had no effect. OV treatment of serum- and glucose-restricted HCT-15 and HCT-116 cells showed no significant increase in the respective oncolytic activities of MeV.

Low protein intake is associated with a major decrease in levels of insulin-like growth factor-1 (IGF-1) [39], which we mimicked by serum restriction in the cell culture medium. Since the insulin-like growth factor receptor (IGF-1R) is overexpressed in different cancer cells [40,41,42] and the IGF-1-dependent pathway plays a major role in cancer cell proliferation, novel approaches aim to inhibit IGF-1R in cancer treatment [43]. Beyond that, OV replication has shown to rely on distinct pathways, e.g., poxvirus JX-594 replication was demonstrated to be limited in cells with activated EGFR/Ras signaling pathways [36,44]. These data suggest that oncolytic viruses may take advantage of starvation-driven alterations in the IGF-1 pathway.

Specifically for MeV, it has been shown that MeV infection exploits autophagy for an optimal replication [45,46]. Normally, autophagy is a cell defense mechanism that activates the lysosomal pathway to fight intracellular pathogens, such as bacteria or viruses. Interestingly, MeV uses autophagy receptors, particularly NDP52 and T6BP, for replication during the course of infection [47]. Since autophagy is upregulated in response to extracellular or intracellular stress, such as starvation, growth factor deprivation, and pathogen infection, a link between autophagy and starvation has been demonstrated [48,49,50]. Since MeV replication depends on autophagy that is being upregulated by starvation, this phenomenon might contribute to MeV-mediated oncolysis.

To verify the mechanisms of increased oncolysis of MeV-GFP in HT-29 CRC cells, we assessed viral replication under starvation conditions. When these CRC cells were fasted 24 h pre-infection and medium was switched to standard medium after infection, viral replication was found to be increased for low-glucose, low-serum levels. A different picture emerged when cells were also fasted after infection. Viral replication in long-term starved cells was diminished by up to 2-log under glucose and serum restriction, but only slightly decreased under serum restriction and standard glucose levels.

A recent study of HSV replication in GBM cells [31] revealed that transient fasting before, but not after, infection raised the yield of HSV both in vitro and in vivo. Remarkably, this effect appeared specifically in cancer cells as there was no increase of viral replication in normal astrocytes [31]. These findings suggest the necessity of sufficient nutrient supply after infection for optimal virus replication, and led to the assumption that the switch from low to high nutrient availability favors cell metabolism for viral replication. This may be explained by the induction of autophagy following short-term starvation, favoring MeV replication [50]. Serum restriction alone seems to have only little impact on viral replication. Moreover, cell cytotoxicity was improved due to increased HSV replication. However, in our study, increased viral replication did not result in enhanced oncolytic activity. In fact, we observed the most significant enhancement of oncolysis under long-term standard glucose, low-serum starvation, where viral replication properties were—even if only slightly—impaired compared to standard cell culture conditions. Therefore, it can be assumed that improved oncolysis in the HT-29 colon carcinoma cell line may be due to mechanisms other than increased viral replication. Though, it would be of interest to further evaluate whether oncolytic vectors in other tumor entities could profit from increased replication in terms of improved cell killing.

Further on, we investigated whether the model of DSR [6] also applies for starvation-based virotherapy. In this study, serum restriction evoked a differential response to virotherapy in normal and neoplastic colon cells. Our data show that long-term serum starvation increased MeV-GFP cell-killing efficacy in the malignant CRC cell line HT-29, whereas it decreased (CCD-18 Co) or was unaffected (CCD-841 CoN) in normal colon cells. Thus, we provide new experimental evidence that differential stress resistance (DSR) plays a pivotal role in the differential biological response of normal and cancerous cells not only to chemotherapy but also to virotherapy.

Note that we used cell line-adjusted MOIs, since the range of oncolysis differs significantly (from 2% to 90% at an MOI of 1) when using the same MOI for the different cell lines of the NCI-60 tumor cell panel [30]. Due to these large differences in oncolytic effectiveness, we used “adjusted”/cell line-specific MOIs for all further experiments to allow enough space to detect the additional cell mass reduction caused by starvation and the combination of both. Otherwise, MeV infection alone could lead to an almost complete destruction of the target cell mass, whereby additional cytotoxic effects possibly caused by our different starvation conditions could no longer be observed. However, the use of different MOIs makes the comparison between different cell lines more difficult, since the availability of glucose or serum per virus particle differs in this setup.

Moreover, we used slightly different standard media containing either 4.5 g glucose/L for malignant cell lines or 1 glucose/L for normal colon cell lines. This leads to a lower availability of glucose per MeV virus particle specifically in the normal cells CCD-18 Co and CCD-841 CoN, and impairs comparability to malignant cell lines. However, cancer cells consume more glucose when cultured in vitro due to their faster proliferation which, to some extent, compensates for the different glucose levels that were applied.

Most tumor cells obtain their increased energy demand from aerobic glycolysis, a significantly more inefficient way to generate ATP than through mitochondrial oxidative phosphorylation, the main ATP source for normal differentiated cells, which is commonly known as the Warburg effect [51,52]. In tumorigenesis, degenerated cells often undergo metabolic stress due to high proliferation rates, insufficient vascularization, and high overall energy consumption. As a consequence of hyperactivation of metabolic pathways, byproducts such as lactate during glycolysis or ROS during oxidative phosphorylation can accrue and damage the tumor cell [53]. Thus, tumor cells counter their increased nutritional demand through broadening the spectrum of energy sources by deriving ATP from fatty acid oxidation (FAO) [54]. In addition, tumor cells can also utilize glutamine or one-carbon amino acids [55,56]. These data suggest that starvation of glucose alone might not be exceedingly detrimental for tumor cells since they are able to reprogram their metabolic needs. This supports our observations that the combination of serum and glucose starvation reduced cell masses significantly compared to glucose starvation alone.

The PI3K/Akt/mTOR pathway and its downstream effectors S6K1 and 4E-BP1 are known to regulate protein synthesis, proliferation, differentiation, and cell survival in response to growth factors, nutrients, and stress [57]. High mTOR activity is common in various cancers, including CRC [58]. Data from experimental studies indicate that features of elevated phosphorylation of Akt and S6K1 in starved breast cancer cells might constitute a possible mechanism of sensitizing cancer cells to chemotherapy [22]. Downregulation of the mTOR pathway due to fasting, on the other hand, may be a contributing factor to the protection of normal cells [31,59,60]. mTOR inhibitors, such as rapamycin, can be used to mimic nutritional depletion in the cell. The combination of rapamycin with oncolytic VACV JX-594 promoted oncolysis and enhanced viral replication in glioma cells, suggesting a critical role of the mTOR pathway for the effectiveness of OVs [61]. Myxoma virus, a rabbit-specific poxvirus with tropism for human cancer cells, has evolved mechanisms to regulate the Akt pathway to establish an environment for optimal viral replication [62]. Since viral vectors rely on host cell metabolism, we speculate that starvation-induced modification of the nutrient-sensing PI3K/Akt/mTOR pathway may also alter the cellular environment in HT-29 cells, and may potentially render cancer cells more susceptible to MeV infection. However, mechanisms which prevent synergistic effects between starvation and virus-mediated oncolysis in other cells, such as HCT-15 and HCT-116, need to be unraveled in further investigations.

Both glucose and serum starvation alone were shown to have significant cytoreductive capacities. The detection of relatively unaffected cell lysis of HT-29 cells following long-term serum starvation without infection support the assumption that cell mass reduction is predominantly mediated through proliferation inhibition rather than cell lysis. These findings match with observations that starvation shifts the cell cycle of pancreatic cancer cells from S phase into G1/G0 phase and thus hinders DNA replication [63].

In future, in vivo studies (both preclinical as well as clinical studies) are required to investigate the combination of starvation and virotherapy in order to determine optimal length of fasting, oncolytic activity, viral replication, and optimal administration of OVs (local vs. systemic treatment). A practical approach for an animal study could be to compare tumor-bearing mice treated with MeV that were either short-term- or long-term-starved with mice fed an ad lib diet. Interesting endpoints would be overall survival, weight development, and (starvation-associated) mortality. Furthermore, it would be of interest whether healthy mice respond differentially to starvation regarding weight loss and mortality when compared to tumor-bearing mice.

Fasting in patients, however, is a matter of debate. While it has been demonstrated to be well-tolerated in most patients with only mild side effects, such as weakness and short-term weight loss in humans [7], it is certainly not suitable for patients with cachexia, sarcopenia, and malnutrition [64,65]. Current guidelines recommend increasing the intake of protein and fat in patients with cachexia [66]. On the other hand, fasting is easily conductible, able to reduce side effects, and potentially effective in a wide variety of tumors, although tumors with PI3K mutations might not be sensitive [67]. Thus, fasting may be limited to relatively stable patients either as a (neo)adjuvant therapy or as a chemotherapy-free approach.

We conclude that starvation-based virotherapy could provide benefits for distinct cancer patients in terms of a personalized starvation-enhanced virotherapy. Despite the fact that fasting requires a lot of perseverance on the patient’s side, most cancer patients have demonstrated a high motivation with regard to gaining the ability to personally contribute in the fight against their malignant disease.

## Figures and Tables

**Figure 1 viruses-11-00614-f001:**
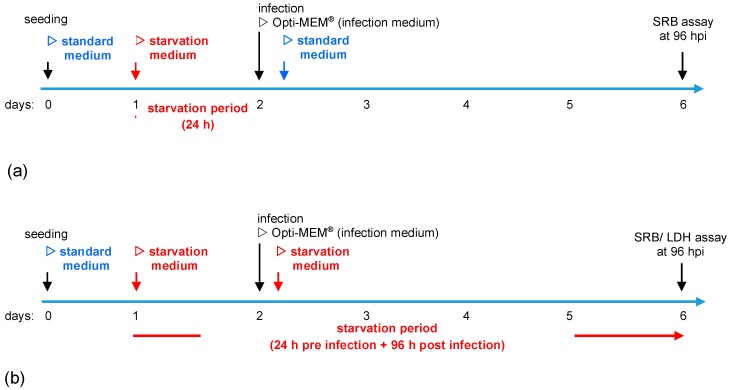
Illustration of starvation conditions. (**a**) Short-term starvation: Cells were seeded in standard medium. On day 1, medium was changed to starvation medium (i.e., starvation medium with variations in contents of glucose and fetal calf serum (FCS)). On day 2, infection with MeV was performed in infection medium (Opti-MEM^®^). At 3 h post-infection (hpi), infection medium was replaced with standard medium. At 96 hpi, the remaining tumor cell masses were determined by the sulforhodamine B (SRB) assay. (**b**) Long-term starvation: Cells were seeded in standard medium on day 0. Medium was changed to starvation medium on day 1. The 3 hpi, infection medium was replaced with the respective starvation medium. At 96 hpi, the remaining tumor cell masses were determined by SRB or lactate dehydrogenase (LDH) assay.

**Figure 2 viruses-11-00614-f002:**
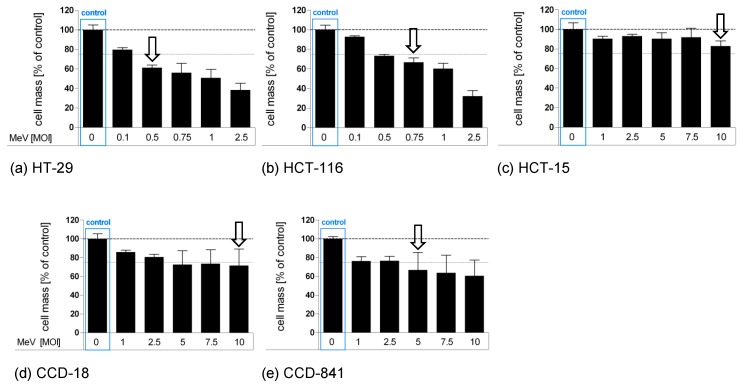
Susceptibilities of human colon carcinoma and normal human colon cells to MeV-mediated oncolysis. Human colon carcinoma cell lines HT-29 (**a**), HCT-116 (**b**), and HCT-15 (**c**) and normal colon cell lines CCD-18 Co (**d**) and CCD-841 CoN (**e**) were cultured in standard medium and infected 48 h after seeding in Opti-MEM^®^ (infection medium) with ascending multiplicities of infection (MOIs as indicated above) of MeV, or remained uninfected (control). At 96 h post-infection (hpi), the remaining tumor cell masses were determined by SRB viability assay. White arrows indicate selected MOIs which were used in further experiments to investigate a combined starvation plus virus-induced tumor cell toxicity. Means and standard error of the means (SEM) of two independent experiments are shown.

**Figure 3 viruses-11-00614-f003:**
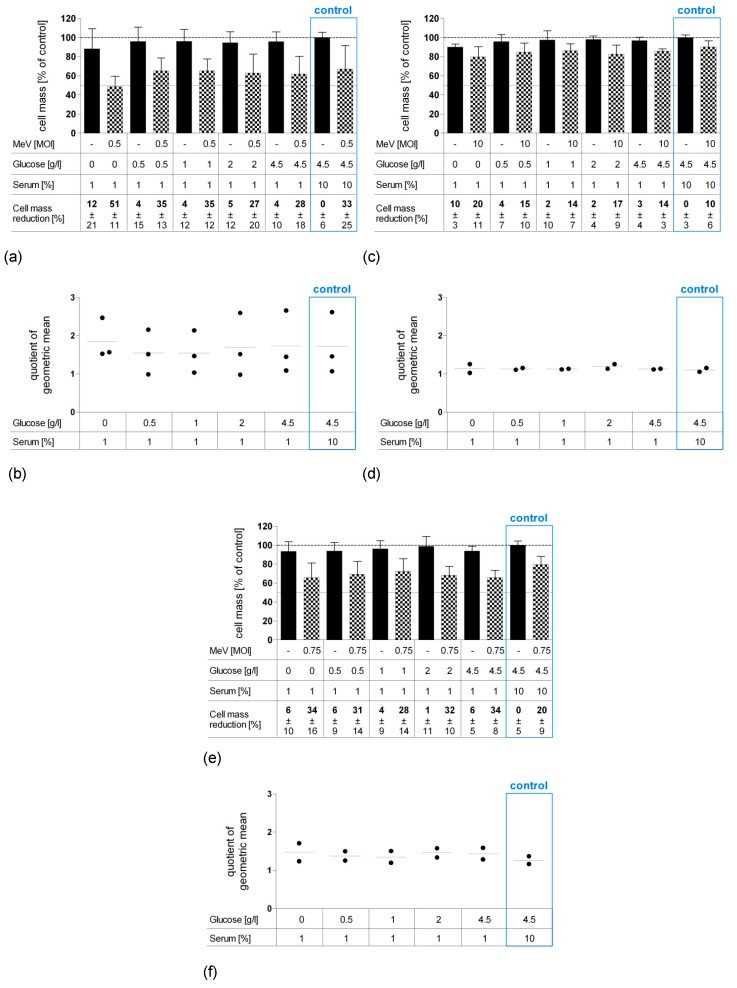
Effect of short-term pre-infection starvation (24 h) on MeV-mediated oncolysis in HT-29 (**a**,**b**), HCT-15 (**c**,**d**) and HCT-116 (**e**,**f**) cells. Short-term starvation and MeV infection of neoplastic HT-29, HCT-15, and HCT-116 cells (human colon carcinoma) were performed according to Figure 1a. At 96 hpi, the remaining tumor cell masses were determined by SRB assay. (**a**,**c**,**e**) Black bars represent MOCK-infected tumor cells (addition of Opti-MEM only, no infectious virus particles); checkered bars display tumor cells infected with MeV. Means and SEM of three or two independent experiments are shown; control: tumor cell cultures and infections performed under standard medium conditions (no starvation). (**b**,**d**,**f**) The impact of starvation on the virus-mediated oncolysis efficacy was evaluated by comparing the ratios of uninfected (MOCK) and virus-infected groups (VIRUS) between starving and standard medium conditions (control). Statistical analysis was performed using the Dunnett’s multiple comparison test. Each dot represents one run of the experiment; horizontal lines represent means of quotient of geometric mean (QoGM).

**Figure 4 viruses-11-00614-f004:**
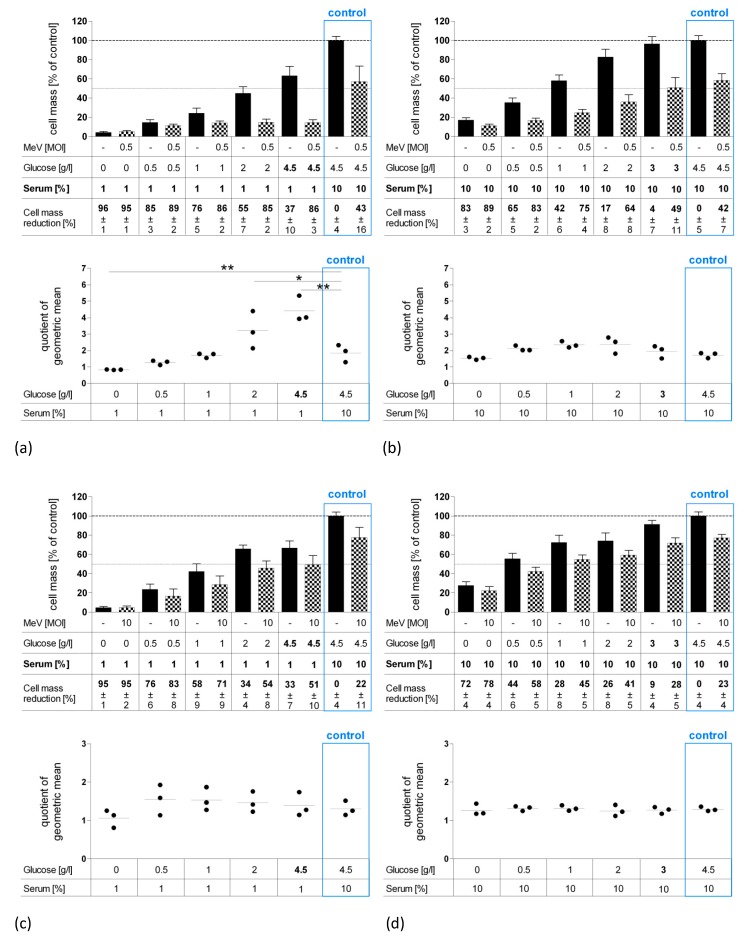
Effect of long-term starvation (120 h) on MeV-mediated oncolysis in HT-29 (**a**,**b**), HCT-15 (**c**,**d**) and HCT-116 (**e**,**f**) cells. Long-term starvation of HT-29, HCT-15 and HCT-116 cells was performed according to Figure 1b. On day 1, medium was changed to low-glucose, low-serum medium (**a**,**c**,**e**) or low-glucose, standard serum medium (**b**,**d**,**f**). At 96 hpi, remaining tumor cell masses were determined by SRB assay. Differences were considered significant when *P*-values were <0.05 (*) or <0.01 (**). Differences in glucose and serum concentrations are pointed out in bold face.

**Figure 5 viruses-11-00614-f005:**
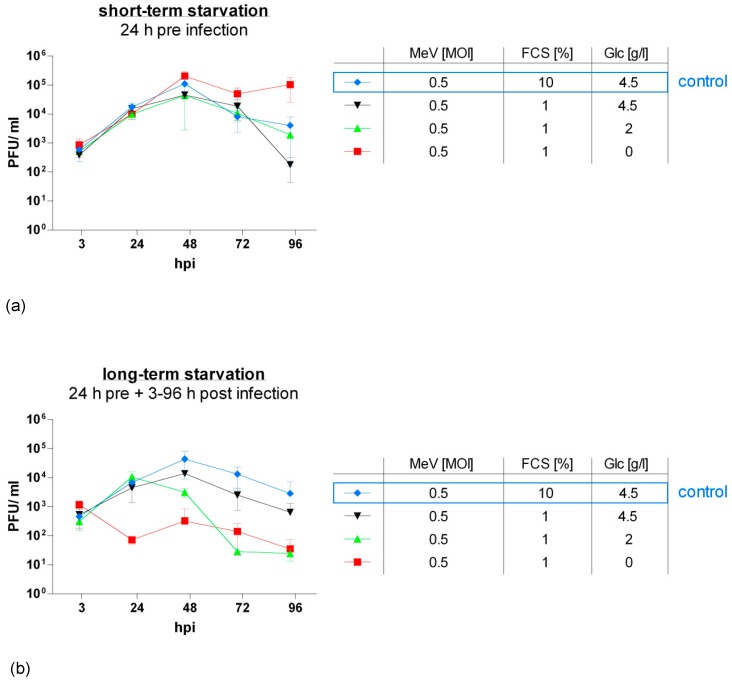
Viral replication under short-term (**a**) or long-term low-serum (**b**) starvation in HT-29 cells. After seeding on day 0, HT-29 cells underwent starvation either for 24 h pre-infection ((**a**): short-term starvation; Figure 1a) or for both 24 h pre- and 3–96 h post-infection ((**b**): long-term starvation; Figure 1b). Infection with MeV (MOI 0.5) was performed on day 2. Supernatants and tumor cell lysates were harvested at 3, 24, 48, 72, and 96 hpi. Titrations were performed on Vero cells and calculated as total amount of plaque-forming units (PFU)/mL (comprising PFU in supernatants plus cell lysates); control: tumor cell cultures and infections performed under standard medium conditions (no starvation). Means and SD of three independent experiments are shown.

**Figure 6 viruses-11-00614-f006:**
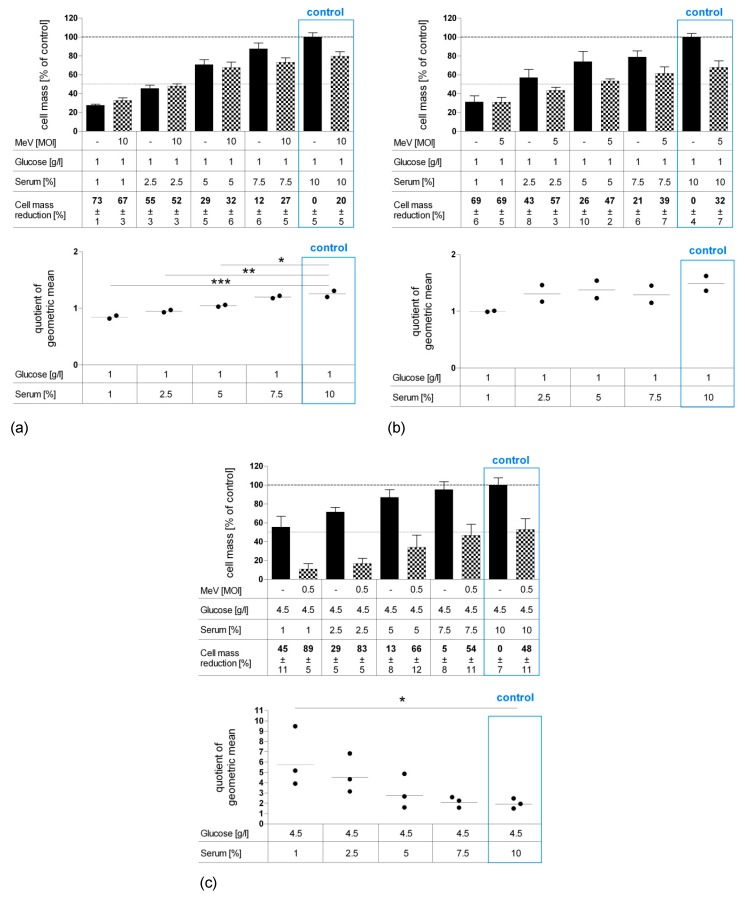
Effect of long-term standard glucose, low-serum starvation on MeV-mediated oncolysis in normal human colon fibroblast cell line CCD-18 Co (**a**) and epithelial cell line CCD-841 CoN (**b**) compared to neoplastic cell line HT-29 (**c**) as determined by SRB assay. Long-term starvation and MeV infection of HT-29, CCD-18, and CCD-841 cells were performed according to Figure 1b. Starvation medium contained standard glucose, but low serum. Note, standard medium in normal colon cells contained only 1 g Glc/L, whereas standard medium in neoplastic cell lines contained 4.5 g Glc/L. For non-malignant cell lines, much higher MOIs were used (MOI 10 and 5) compared to the malignant cell line (MOI 0.5). At 96 hpi, the remaining tumor cell masses were determined by SRB assay. Differences were considered significant when *P*-values were <0.05 (*), <0.01 (**), and <0.0001 (***).

**Figure 7 viruses-11-00614-f007:**
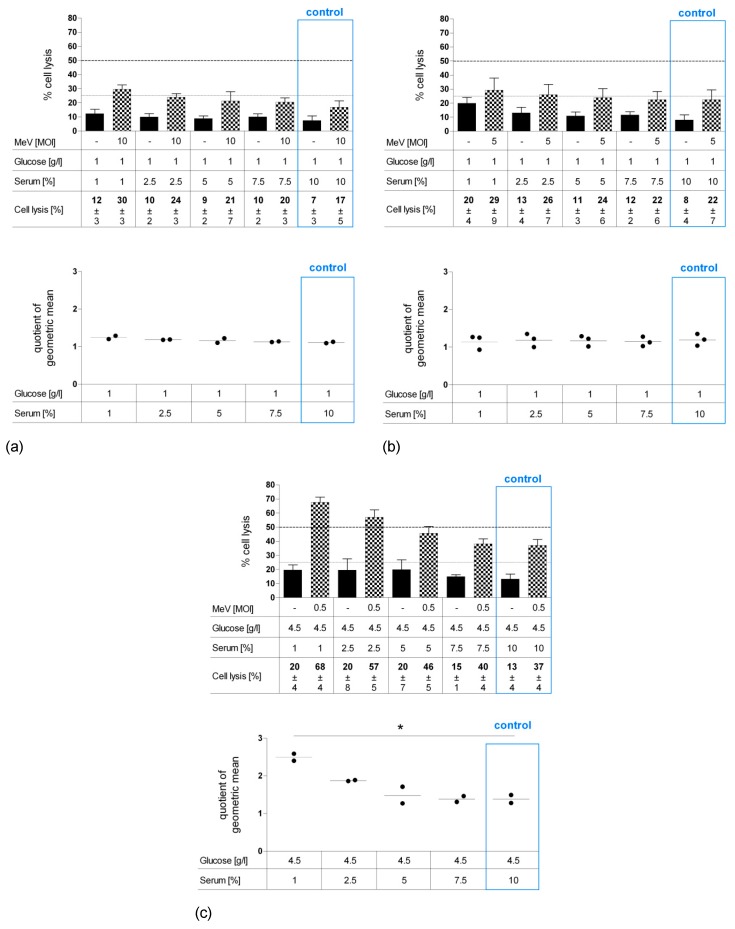
Effect of long-term standard glucose, low-serum starvation on MeV-mediated oncolysis in normal human colon fibroblast cell line CCD-18 Co (**a**) and epithelial cell line CCD-841 CoN (**b**) compared to HT-29 cells (**c**) determined by LDH assay. Cell culture, starvation and infection were carried out as in Figure 6. At 96 hpi, an LDH assay was performed to determine cell lysis. Differences were considered significant when *P*-values were <0.05 (*).

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
