# Peer review of "Starvation-Induced Differential Virotherapy Using an Oncolytic Measles Vaccine Virus"

_viruses, 2019, doi:10.3390/v11070614_

Reviewer 1 Report

The paper explores the effects of glucose and serum starvation on OV replication in a limited number of cell lines.  Data interpretation would be strengthened if more cell lines were compared.

Abstract

Lines 24-27.  These lines are difficult to read - they sound contradictory even though they aren't.  I recommend that you standardize the terminology for the different culture media conditions that were used.  Maybe avoid the term 'starved' and specify if glucose concentration was reduced or a standard (growth) media concentration was used.  Likewise, clearly indicate if serum concentration was reduced or a standard (growth) media concentration was used.

Introduction

Lines 35-36. Please add a reference.

Line 44.  "...caloric restriction which beyond promotes chronic weight loss..." is poorly worded.  Please modify the sentence.

Line 53.  "...starvation of tumor-bearing mice allowed to increase the..." seems to be missing a subject.  Maybe try "...starvation of tumor-bearing mice allowed researchers to increase the..."?

Line 54.  The human dose is irrelevant here.  How did mice without tumors fare?

Line 64.  Reword to "...after administration is being tested in patients..."

Lines 74-77.  The beginning of this sentence is generalized to all OVs but only one reference is provided.  Please add references for other OVs or reword the sentence.  Suggested change:  "However, widespread resistance to oncolysis following MeV inoculation was observed using the NCI-60 tumor cell panel..."

Lines 81-83.  I'm not sure using DSS and DSR is necessary.  It is the same mechanism, it's just that healthy and cancer cells respond differently.  Consider rewording and being more true to the definition of DSR.

Materials & Methods

Lines 145-153.  Specifically tell the reader what SRB detects.

Lines 154-163.  Specifically tell the reader what LDH tells you.

Results

Lines 203-204.  I'm not sure the individualized cell line MOI adjustments were necessary or appropriate.  Do you have data at the same MOI for each cell line too?

Line 226.  Explain how using a vaccine with GFP in it is "state-of-the-art".

Lines 228-230.  An MOI = 10 is probably higher than you could ever reach in a cancer patient too.

Lines 225-232.  Point out that these cells are the neoplastic cells.

Lines 236-238.  Please show this data or rephrase this statement to make it more hypothetical.

Line 244.  Change "blue boxes".  No color is shown.

Lines 274-275.  Data for HCT 116 does not support this statement.  Be more careful about interpretation of the limited data you have.

Lines 275-277.  It would be nice to see all cell lines in the Figures (rather than making some data supplementary information).  You would not need to repeat data shown in the Figures as text that way.

Lines 279-280.  Consider rewording "...not found to be suppressed under starvation...".  It's not really what you're asking.  You are asking if starvation potentiates virus replication in cancer cells.

Lines 307-310.  Check your numbers.  I don't think they are reflected in the data you show.  It would be nice to see all cell lines in the Figures (rather than making some data supplementary information).  You would not need to repeat data shown in the Figures as text that way.

Line 321.  Reword "oncolysis rate".  Percentages are not rates of lysis.

Lines 328-329.  The summary statement is overstated.  Your data only illustrate this in one cell line.  Data from one cell line are not enough information to make this conclusion.

Lines 333-335.  Please explain how you reached this conclusion.

Lines 336-337.  Keep terms the same throughout.  Change "short-term starved" to "low glucose, low serum media"?

Lines 358-364.  You don't have to repeat data shown in the Figure.

Lines 365-366.  Remove this sentence.  It's not supported by the data.

Line 392.  Change to "At low glucose concentration (1 g/L), lower concentrations of serum led to reduction of cell mass."  Indicate if this was a significant reduction or not.  If Fig 5c does not use the same glucose concentrations, you cannot compare the data directly.

Lines 398-400.  Indicate why it is important.

Be careful about putting too much discussion in the results section.

Discussion

Line 427.  Change "was" to "has been".

Lines 431-433.  Suggested change:  "Effective OVs can selectively... ...OVs may initiate a profound and..."

Line 437.  Delete "state-of-the-art".

Line 446.  Discuss results in the other cells tested too.

Lines 448-455.  What has been shown with MeV?

Line 473.  Don't generalize.  Suggested change:  "...oncolysis in the HT-29 colon carcinoma cell line..."

Line 478.  Add reference re vulnerability to stress.

Lines 478-484.  This gets somewhat repetitive and can be combined with earlier sections of the discussion.

Line 507-512.  Different viruses respond differently.  I don't think you can assume MeV requires the same pathways that poxviruses do.  Add references re myxoma virus and the mTOR pathway.

Line 513.  Suggested change:  "Combination glucose and serum starvation proved to..."

Figures

Supplementary Figure 1 would be beneficial to include in the paper (not as a supplement).

Figure 2.  Are any of these differences significant?

Figure 3.  Point out the differences in the glucose concentrations in (a) and (b) [4.5 vs. 3].

Figure 5.  Should the glucose concentration in 5c be 1 g/L?  Change the figure legend to indicate only one malignant cell line was tested (Line 388).

Reviewer 2 Report

The manuscript written by Scheubeck et al reports that the starvation-based virotherapy increased the oncolytic potential of oncolytic measles vaccine virus in colon cancer patients.

1) Introduction: authors should also report the use of GMCSF encoding 3/5 chimeric adenovirus for the treatment of mesothelioma and reovirus that are now evaluated in clinical trials

2) The effect of starvation should have been observed also by evaluating the apoptotic and necrotic cell death

2) This study lucks of in vivo data. In vivo efficacy studies confirming the results observed in vitro would be helpful

Author Response

Reviewer #2 (Comments to the Authors)

The manuscript written by Scheubeck et al. reports that the starvation-based virotherapy increased the oncolytic potential of oncolytic measles vaccine virus in colon cancer patients.

1)  Introduction: authors should also report the use of GM-CSF-encoding 3/5 chimeric adenovirus for the treatment of mesothelioma and reovirus that are now evaluated in clinical trials.

Our response:  Thank you for the valuable input. We included a section about the use of GM-CSF-encoding viruses, such as adenovirus and reovirus, that are being evaluated in clinical trials.

Accordingly, our manuscript now has been supplemented as follows (line 91 - 101):

ONCOS-102, a GM-CSF encoding oncolytic adenovirus, has demonstrated to induce infiltration of CD8+ T-lymphocytes into initially T-cell negative mesothelioma in a 68 year-old patient [16]. Accordingly, also poorly immunogenic tumors could be sensitized to immunotherapy by OVs which opens up new possibilities for T-cell based approaches in cancer treatment.

Reovirus is another OV being evaluated in numerous tumor entities in clinical trials in combination with conventional chemotherapy [16-21]. REOLYSIN®, an isolate of reovirus type 3, was well tolerated in patients, but did not improve progression-free survival (PFS) in metastatic colorectal carcinoma, NSCLC, breast cancer and prostate cancer compared to standard treatment arms; however, it proved to extend overall specific survival specifically in breast cancer patients [16, 17, 20, 21]. Reovirus is currently investigated in a Phase I trial in combination with GM-CSF for the treatment of high-grade relapsed or refractory brain tumors (NCT02444546).

2) The effect of starvation should have been observed also by evaluating the apoptotic and necrotic cell death

Our response:  Admittedly, this information would have been beneficial. We know from data reported by Xia et al., 2014, that oncolytic MeV utilizes mitophagy to counteract apoptosis in NSCLC cells favoring persistent viral replication which finally results in necrotic cell death due to ATP exhaustion.

Ø  Xia, M., et al., Mitophagy switches cell death from apoptosis to necrosis in NSCLC cells treated with oncolytic measles virus. Oncotarget, 2014. 5(11): p. 3907-18.

Based on these results, we postulate, cell death caused by MeV is mainly due to necrosis. This hypothesis is supported by our data from the LDH-assay, shown in Figure 5: MeV induced more than twice as much LDH release than starvation. We conclude that MeV mainly induces necrotic cell death whereas starvation retards tumor cell growth mostly via growth inhibition rather than apoptosis or necrosis. Unfortunately, based on our data, we cannot differentiate between apoptotic and necrotic cell death caused by starvation or virotherapy, respectively, and more experiments are needed to unravel the exact mechanisms triggered by starvation.

3)  This study lacks of in vivo data. In vivo efficacy studies confirming the results observed in vitro would be helpful 

Our response: We confirm that there is a lack of in vivo data which would strengthen our interpretation. Until now, there is only one study reporting in vivo data of starvation-based virotherapy.

This study by Esaki et al., 2016, showed in a small number of mice (n=8) with orthotopic glioblastoma multiforme xenografts, that replication of oHSV was significantly enhanced in tumors after starvation for 48 h followed by 24 h of normal diet (n=4) before infection compared to mice fed ad lib diet (n=4).

Ø  Esaki, S.; Rabkin, S. D.; Martuza, R. L.; Wakimoto, H., Transient fasting enhances replication of oncolytic herpes simplex virus in glioblastoma. Am. J. Cancer Res. 2016, 6, (2), 300-11.

This data matches with our observations, that MeV replication was increased after intense short-term low glucose, low serum starvation in HT-29 cells.

Further in vivo efficacy studies are required to investigate toxicity, PFS and overall survival (OS) of starvation-based virotherapy. We included a section addressing this matter in the discussion.

Nevertheless, we do hope that you consider this article for publication despite the fact that at this stage of our research program in vivo data are still lacking.

The revised version of our manuscript now has been changed as follows (line 702 - 709):

Further on, in vivo studies (both preclinical as well as clinical studies) are required to investigate the combination of starvation and virotherapy in order to determine optimal length of fasting, oncolytic activity, viral replication and optimal administration of OVs (local vs. systemic treatment). A practical approach for an animal study could be to compare tumor bearing mice treated with MeV that were either short-term or long-term starved with mice fed ad lib diet. Interesting end points would be overall survival, weight development and (starvation-associated) mortality. Furthermore, it would be of interest, whether healthy mice respond differentially to starvation compared to tumor bearing mice regarding weight loss and mortality.

Reviewer 3 Report

1. In this study, the authors indicated that long-term starvation is capable of enhancing the oncolytic potential of MeV-GFP specifically in the human colon cancer cell line HT-29. An important question is that can this finding be used in cancer patient treatment. It is impossible to have patients fasting for more than a few days. Even so, cancer cells are still capable to obtain their nutrition by taking advantages of normal cells and tissues. Therefore, how this finding with cells in culture could practically be tested in animals and further potentially applied in cancer patients. This only briefly discussed at the end of paper. It would be nice to have more discussion.

2. Relatively to above comment, in the introduction, many references are cited. It would be better also to include some of key results of the most relative studies, such as the clinical trials of fasting (if any available) and the reference 21.

3. In text regarding figure 5, normal cells, CCD-18 Co (Figure 5a) and CCD-841 CoN (Figure 5b), and cancer cell line HT-29 (Figure 5c) cells were treated long-term standard glucose, low serum starvation. However, in the figure 5, normal cells CCD-18 and CCD-841 were treated with low glucose (1%), while HT-29 (Fig. 5c) was treated with 4.5% glucose). It seems that the standard glucose for normal cells is 1%, but for HT-29 is 4.5%. If so, it should indicate in there (again) to make it clearer.

Author Response

Reviewer #3 (Comments to the Authors)

1) In this study, the authors indicated that long-term starvation is capable of enhancing the oncolytic potential of MeV-GFP specifically in the human colon cancer cell line HT-29. An important question is that can this finding be used in cancer patient treatment. It is impossible to have patients fasting for more than a few days. Even so, cancer cells are still capable to obtain their nutrition by taking advantages of normal cells and tissues. Therefore, how this finding with cells in culture could practically be tested in animals and further potentially applied in cancer patients. This only briefly discussed at the end of paper. It would be nice to have more discussion. 

Our response:  We fully agree with you that the discussion about how these findings can be translated into the clinic practice is crucial for the value of this study. We followed your advice and included specific paragraphs addressing this important matter in the discussion section.

Accordingly, the revised version of our manuscript now has been changed as follows (line 702-717):

Further on, in vivo studies (both preclinical as well as clinical studies) are required to investigate the combination of starvation and virotherapy in order to determine optimal length of fasting, oncolytic activity, viral replication and optimal administration of OVs (local vs. systemic treatment). A practical approach for an animal study could be to compare tumor bearing mice treated with MeV that were either short-term or long-term starved with mice fed ad lib diet. Interesting end points would be overall survival, weight development and (starvation-associated) mortality. Furthermore, it would be of interest, whether healthy mice respond differentially to starvation compared to tumor bearing mice regarding weight loss and mortality. 

Fasting in patients however is a matter of debate. While it has been proven to be well tolerated in most patients with only mild side effects such as weakness and short-term weight loss [3], it is certainly not suitable for patients with significant stages of cachexia, sarcopenia and malnutrition [64, 65]. Current guidelines recommend to increase the intake of protein and fat in patients with cachexia [66]. On the other hand, fasting is easily conductible, able to reduce side effects and potentially effective in a wide variety of tumors. Thus, fasting may be limited to relatively stable patients either as a (neo)adjuvant therapy or as a chemotherapy-free approach.

2)  Relatively to above comment, in the introduction, many references are cited. It would be better also to include some of the key results of the most relative studies, such as the clinical trials of fasting (if any available) and the reference 21.

Our response:  We highly appreciate your advice. Since we are physicians, results from clinical trials are indeed highly relevant to us. We included all relevant studies published about combined fasting and chemotherapy in cancer patients.

Furthermore, we added NCT numbers of all ongoing clinical trials.

We furthermore included a short summary of the key results of the reference 21.

Accordingly, the revised version of our manuscript now has been changed as follows (line 57 - 73):

To date, only a small number of clinical trials have explored the effect of combined fasting and chemo­therapy in patients [3, 8-10]. A case report on 10 patients suffering from different cancer entities showed a significant reduction of side effects such as fatigue or weakness when fasting was performed 48-140 h prior to and 5-56 h post chemotherapy [3]. Another more recent clinical study evaluated the effects of short-term fasting on tolerance to adjuvant chemotherapy in HER-2 negative breast cancer patients [9]. As a result, erythrocyte and thrombocyte counts post chemotherapy were higher in fasted patients. In 34 women with breast and ovarian cancer, 60 h fasting plus chemo­therapy not only proved to be safe and feasible, but also to improve quality of life, well-being and fatigue compared to chemotherapy alone [10]. Accordingly, fasting was well tolerated in cancer patients and has the potential to reduce side effects, but lacks data supporting that it can increase the efficiency of current anticancer therapies, as data from in vitro and animal studies has demonstrated.

Many larger clinical trials are currently ongoing to determine possible benefits of fasting regarding efficacy of treatment, adverse events, quality of life, weight changes, and changes in metabolic, hormone, and inflammatory response (NCT00936364, NCT01802346, NCT02710721, NCT03162289, NCT03340935, NCT03595540, NCT03709147, NCT03700437, NCT01175837; please note: cited clinical studies are denoted by their ClinicalTrials.gov identifiers).

Line 108 - 112:

Until now, only one study was published by Esaki et al. investigating whether starvation may enhance the effectiveness of oncolytic viruses (OVs) [31]. In this study, fasting demonstrated to increase the replication and oncolytic activity of oncolytic herpes simplex virus (oHSV) in glioblastoma multiforme (GBM) cells, but not in human astrocytes. These results were confirmed in vivo, showing enhanced virus replication in starved mice [31].

3)  In text regarding figure 5, normal cells, CCD-18 Co (Figure 5a) and CCD-841 CoN (Figure 5b), and cancer cell line HT-29 (Figure 5c) cells were treated long-term standard glucose, low serum starvation. However, in the figure 5, normal cells CCD-18 and CCD-841 were treated with low glucose (1%), while HT-29 (Fig. 5c) was treated with 4.5% glucose). It seems that the standard glucose for normal cells is 1%, but for HT-29 is 4.5%. If so, it should indicate in there (again) to make it clearer. 

Our response:  You are totally right. In case of normal human colon cells, Alpha MEM (containing 1 g Glc/l) is the standard medium for cell culture. For neoplastic colon cells the standard medium is DMEM which contains 4.5 g Glc/l (see materials & methods).

We addressed your comment by outlining the differences between the standard growth media for malignant and non-malignant cell lines in the results section regarding figure 6.

Accordingly, the revised version of our manuscript now has been changed as follows (line 519 - 523):

“Starvation only” reduced cell masses in non-malignant cells CCD-18 Co (Figure 6a) and in non-malignant CCD-841 CoN cells (Figure 6b) at descending FCS concentrations. In malignant HT-29 cells, lower concentrations of serum led to reduction of cell mass (Figure 6c). Note, standard medium for normal colon cells contains only 1 g Glc/l, whereas standard medium for neoplastic cells contains 4.5 g Glc/l.

Round  2

Reviewer 1 Report

Please include a discussion of the complications with comparing data from the different cell lines given that different definitions of 'standard media' and different moi's were used.

Author Response

Please include a discussion of the complications with comparing data from the different cell lines given that different definitions of 'standard media' and different moi's were used.

Our response:  We included a section about the complications that go along with using different definitions of “standard media” and different MOIs.

Accordingly, the revised version of our manuscript now has been changed as follows (line 514 - 528):

Please note, we used cell line adjusted MOIs, since the range of oncolysis differs significantly (from 2 % - > 90% at an MOI of 1) when using the same MOI for the different cell lines of the NCI-60 tumor cell panel [30]. Due to these large differences in oncolytic effectiveness, from then on, we used “adjusted”/cell line specific MOIs to allow enough space to detect the additional cell mass reduction caused by starvation and the combination of both. Otherwise, oncolysis per se already could have achieved a near total destruction of the target cell mass thereby not allowing to observe any add on/additional cytotoxic effects caused by our different conditions of starvation. However, the use of different MOIs makes the comparison between different cell lines more difficult, since the availability of glucose or serum per virus particle differs in this setup.

Moreover, we used slightly different standard media containing either 4.5 g glucose/l for malignant cell lines or 1 glucose/l for normal colon cell lines. This leads to a lower availability of glucose per MeV virus particle specifically in the normal cells CCD-18 and CCD-841 and impairs comparability to malignant cell lines. However, cancer cells consume more glucose when cultured in vitro due to their faster proliferation which, to some extent, compensates for the different glucose levels that were applied.

Reviewer 2 Report

Authors improved the quality of manuscript which is now suitable for publication

Author Response

Thank you for your positive response.

In response to Revier #1 we included one section in the discussion about the use of different MOIs and different standard media (please see the attachment) in our 2nd revision of the manuscript.

Besides that there has nothing been changed.
